civil engineering

steel tube slab method, large span, construction sequence, ground surface settlement, structural deformation

**Author for correspondence:**
Wen Zhao
e-mail: wenneu@163.com

# Study on ground settlement and structural deformation for large span subway station using a new pre-supporting system

Pengjiao Jia[1], Wen Zhao[1], Xi Du[2], Yang Chen[1], Chaozhe Zhang[1], Qian Bai[1] and Zhiguo Wang[1]

[1]School of Resources and Civil Engineering, Northeastern University, NO. 3-11, Wenhua Road, Heping District, Shenyang, People's Republic of China
[2]School of Civil and Environmental Engineering, UNSW, Sydney, New South Wales 2052, Australia

PJ, 0000-0001-6605-793X; YC, 0000-0003-2730-1965

This study presents a novel construction pre-supporting system for large underground space excavation with shallow depth, Steel Tube Slab system (STS), in which adjacent steel pipes are connected by a couple of flanges, bolts and concrete for flexural capacity and lateral stiffness of the whole structure. The STS method is employed for the first time for the construction of the ultra-shallow buried and large span subway station in China, during which ground settlement and structural deformation are monitored. A numerical model for the subway station is established with reliability verified by monitored data comparison from numerical results and investigation on the effect of large span underground excavation on surrounding soil surrounding soil and existing buildings in soft soils. Unlike traditional methods, the STS method can effectively control and reduce the ground settlement during construction, thereby rendering it ideally suited for application in soft soils.

## 1. Introduction

Over the past few decades, concerns for the stability and safety of adjacent existing buildings and structures in urban areas have warranted studies on underground excavation techniques. While stress redistribution of the surrounding soils around the excavation face can cause great damage without an effective supporting system [1,2], the ground settlement during subway station construction in populous city areas is the most important

factor that needs to be controlled, giving rise to several propositions for ground settlement prediction during underground excavation from several researchers including Peck [3], Moghaddasi & Bidgoli [4] and Chakeri et al. [5].

Typical features such as ultra-shallow buried excavation, complicated upper building environments and large spans in underground excavation projects in urban districts are responded to in the design and construction of underground structures for excavation-induced ground settlement and structural stability [6]. As a result, a large number of new supporting systems and excavation methods for underground construction have been developed, with some listed below.

The New Tubular Roof (NTR) Method is one of the new methods developed for the design and construction of subway stations [7]. The permanent supporting structure is pre-built by forming a connected tube space with a high stiffness steel tube, and then the large span underground space was excavated under the permanent support structure. The NTR method can effectively control the ground settlement because large-caliber steel tubes are jacked into the target position with little disturbance to the surrounding soils [8–10]. This technique has been successfully applied in the construction of Metro line #9 of the Seoul Subway, Xinle Yizhi station of Shengyang Subway and also the Gongbei tunnel [7,11–13]. Unfortunately, the underground structure built by this method is arch-shaped, which has low utilization for underground spaces.

Another new kind of pre-supporting system, referred to as the Concrete Arch Pre-supporting System, was proposed by Dadizadeh and Sadaghiani for construction of subway stations [14]. This method has been used in the construction of the Tehran Metro since 2002 and is a perfect method for construction of large span underground spaces in ultra-shallow-buried ground. In this method, underground reinforced concrete systems containing arch beams and piles are constructed around the designed underground space ahead of the main excavation period so that the surcharge of soil is very large during the underground excavation.

In addition, the Central Beam Column (CBC) system has also been proposed for the construction of the Tehran Subway [15]. The main phases of construction are: (1) excavation and pre-supporting the lower portion of the centre drifts; (2) construction of CBC reinforced concrete structure including the bottom beam, centre columns and top beam; (3) excavation and pre-supporting of side drifts and middle drifts, respectively; (4) construction of the final lining. The CBC supporting system can increase the stiffness of supporting structures and reduce the deformation of soil above the excavation face, but the construction sequences are too complicated to be widely used in engineering construction.

In response to the limitations mentioned above, this paper proposes a novel technique, i.e. STS, as a pre-supporting system for underground excavations, adopted in the Northeastern Street station of line #10 in Shenyang Subway. The ground surface settlement and structural deformation for large span subway station using the STS method were monitored and analysed and the influences of some key parameters such as construction sequence and temporary steel supports on the ground settlement control are investigated. This paper begins with a discussion on techniques for underground excavations and is followed by a description of the STS technique. Numerical analysis of subway station construction using STS structure and an engineering project are presented, along with discussions on future work.

## 2. Description of STS structure

The STS structure originates in the traditional pipe-roofing supporting system. Considering the weak connection in the transverse bearing capacity, the adjacent steel tubes are connected by flange plates, steel bolts and concrete [16–19], as shown in figure 1. This connection significantly improves the transverse stiffness and ultimate bearing capacity of the supporting structure. Complicated temporary steel supports can be avoided during excavation, which can ensure the safety of construction, decrease the construction difficulties and also reduce construction time compared to traditional pipe-roofing systems [20].

In general, the STS structure combined with the PBA method is used to perform underground excavation. The main construction steps are described as follows and the construction sketch diagrams are depicted in figure 2.

(a)  The designed steel tubes are jacked into the soil layer by the mechanical equipment, and then the soil is cleaned both inside and outside of the steel tubes.
(b)  The transverse steel bolts are installed in the specified location, and self-compacting concrete is casted in the tubes by the reserved grouting holes.

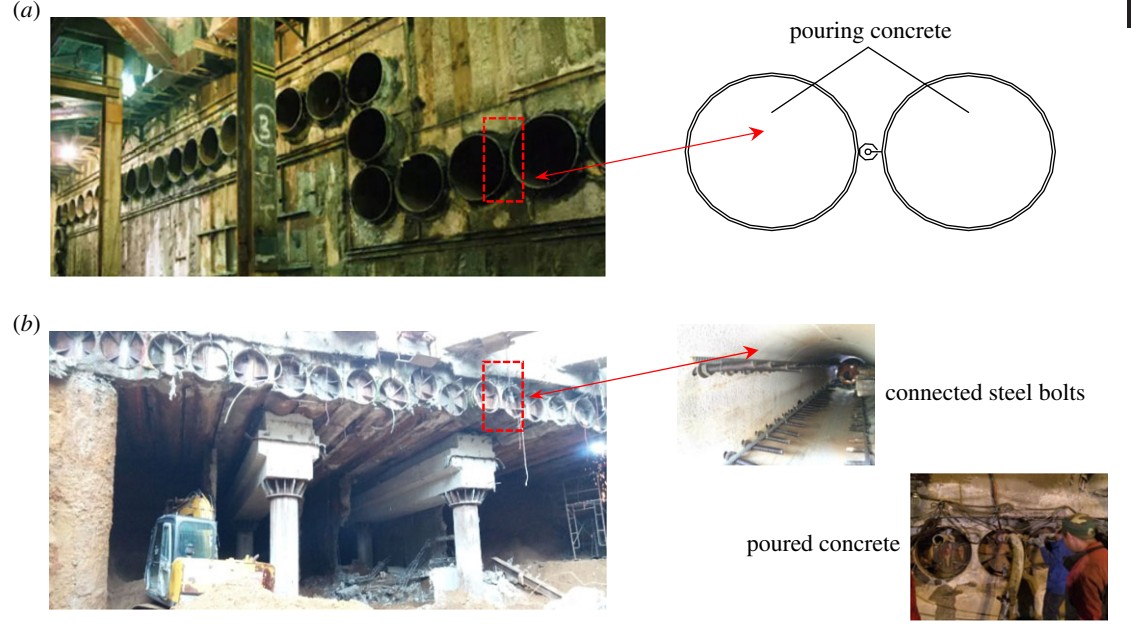

**Figure 1.** The supporting system. (*a*) Traditional pipe-roofing system and (*b*) steel tube slab structure.

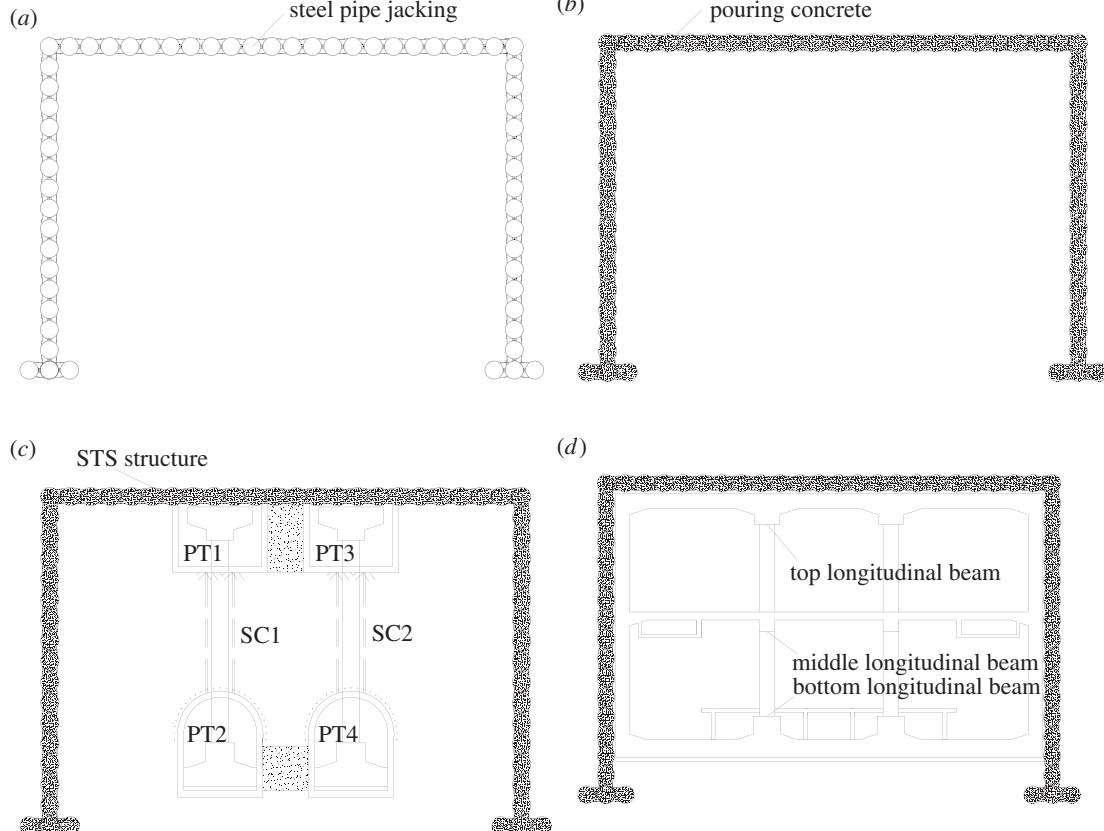

**Figure 2.** The main construction sequences. (*a*) Steel pipe jacking, (*b*) pouring concrete, (*c*) beam-column system construction and (*d*) the main structure construction.

 (c) The pilot tunnels (PT1 and PT2) are constructed first, then the steel column (SC1) is constructed. The rest of the gradients (PTZ3, PT4 and SC2) are completed using identical construction sequences.
 (d) The subway hall and the platform floor, respectively, are constructed.

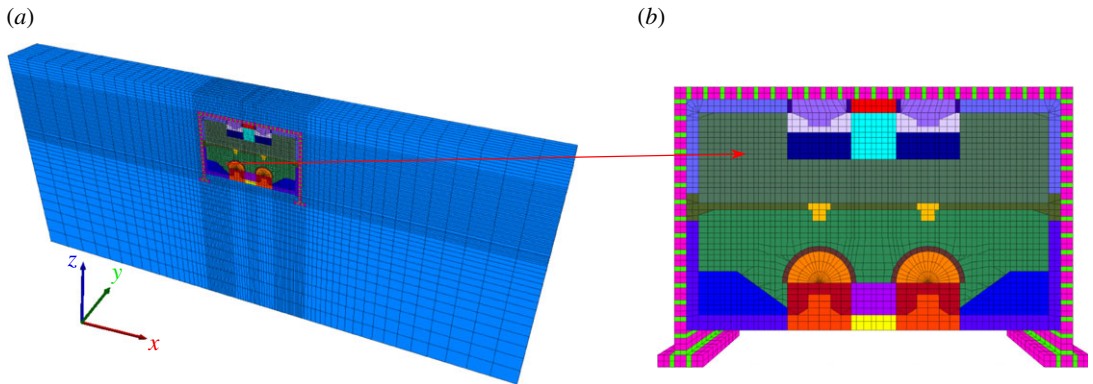

**Figure 3.** Three-dimensional difference model of the station construction. (*a*) The numerical model and (*b*) details of the support structure.

# 3. Numerical analysis of subway station construction using STS structure

There are few studies on STS structure as this method is being used for construction of an ultra-shallow-buried subway station for the first time. A three-dimensional finite-element model is used to investigate the influence of construction sequences on ground settlement and STS structure deformation.

## 3.1. The difference model

### 3.1.1. Basic assumption

(a) The soil was modelled as a linear plastic material yielding according to the Mohr–Coulomb failure criteria. Soil in the same layer has an elastic plastic property which can be treated as homogeneous and isotropic.
(b) The effect of groundwater seepage is not considered because the groundwater drainage is completed before construction.
(c) The elastic model is adopted for the calculation of the structure of the lining and beam-column system, in which elastic deformation is the only factor considered. The weight of the soil and the main structure together with the ground overload are considered.

### 3.1.2. Model parameters

A three-dimensional finite difference model is established to study the surface settlement and structural deformation of the proposed supporting system using FLAC$^{3D}$. The size of the numerical model is $120 \times 9 \times 50$ m, as shown in figure 3. All the directions of the bottom surface of the model are fixed, and the horizontal displacements are only fixed in the left and right sides. The top surface is free to move in all directions. The numerical model is difficult to converge due to the complex connections of the STS structures. An equivalent structure is established to avoid the aforementioned problems which had been listed in existing literature [6]. A surcharge of 20 kPa is applied on the top surface to simulate traffic loads. The diameter of the steel columns is 0.8 m, the distance between adjacent steel columns is 8.5 m, the length of the feet-lock bolt is 2.5 m, and the distance between the feet-lock bolts is 1.0 m in the model.

Beam and pile elements were deployed to simulate the temporary steel supports and feet-lock bolt, respectively. Line and shell elements were applied to simulate the steel columns and the primary lining, respectively. Meanwhile, the Mohr–Coulomb model was adopted to define the behaviour of soil, and the soil properties are shown in table 1. STS structure, primary lining, beam and columns were assumed to have a liner-elastic behaviour, and the properties of the material are listed in table 2.

### 3.1.3. Construction sequences

The main construction simulation was divided into nine stages, as follows:

(1) Stage 1: Construction of the left-bottom pilot tunnel (PT2);
(2) Stage 2: Construction of the left-top pilot tunnel (PT1);

**Table 1.** Soil parameters at the site.

| soil | thickness (m) | internal friction angle (°) | cohesion $c$ (kPa) | Poisson ratio | deformation modulus (MPa) | density (kg m$^{-3}$) |
| --- | --- | --- | --- | --- | --- | --- |
| backfill soil | 3.4 | 10 | 1 | 0.28 | 16 | 1800 |
| gravel sand | 6.0 | 37.0 | 1 | 0.26 | 33.0 | 2000 |
| round gravel | 11.0 | 36.7 | 1 | 0.25 | 30.9 | 2050 |
| medium coarse sand | 10.0 | 34.0 | 1 | 0.29 | 19.0 | 1980 |
| boulder clay | non-penetrating layer | 29.3 | 1 | 0.30 | 23.0 | 2000 |

**Table 2.** Material parameters.

| material | $\mu$ | $E$ (GPa) | $\gamma$ (kN m$^{-3}$) |
| --- | --- | --- | --- |
| primary lining | 0.2 | 20 | 24 |
| foot-lock bolt | 0.25 | 72 | 25 |
| top longitudinal beam | 0.2 | 32.5 | 24 |
| bottom longitudinal beam | 0.2 | 32.5 | 24 |
| steel column | 0.25 | 69 | 38 |

(3) Stage 3: Construction of the left cylindrical steel column (SC1);
(4) Stage 4: Construction of the right-bottom pilot tunnel (PT4);
(5) Stage 5: Construction of the right-top pilot tunnel (PT3);
(6) Stage 6: Construction of the right cylindrical steel column (SC2);
(7) Stage 7: Construction of the top plate at mid-span;
(8) Stage 8: Construction of the station hall; and
(9) Stage 9: Construction of the platform layer.

## 3.2. Numerical analysis of construction process

### 3.2.1. Analysis of ground settlement

Figure 4 shows the ground settlement curve for each construction stage. It is noted that the maximum surface settlement occurs at the end of the excavation of the station hall and the maximum settlement is 7.7 mm. The influence of underground excavation on ground settlement reaches its maximum towards the end of the main structural construction. The centre line of the maximum settlement gradually moves right during the construction of the right pilot tunnel. The buckle arch construction stage has greatly influenced the maximum settlement, increasing by about 3.2 mm. The STS structure has been constructed ahead of the excavation construction so that the main structural excavation has little impact on the surrounding environment, and the main affected area is directly above the station.

It can be seen from figure 4 that the settlement caused by stage 1(PT2) and the construction of the mid-span buckle arch account for 25.4% and 37.5% of the maximum settlement, respectively. The settlement caused by stage 2 (PT1) is relatively small, accounting for only 6.8%. The settlement caused by the construction of the bottom-left pilot tunnel (PT2), the top-left pilot tunnel (PT1) and the left column (SC1) accounted for 43.5% of the total settlement. The settlement caused by the construction of the bottom-right pilot tunnel (PT4), the top-right pilot tunnel (PT3) and the right column (SC2) accounted for 15.2% of the total settlement. The reason is that after the construction of the left part of the structure, the overall cross section was reduced, resulting in less impact on the total settlement when the right part was being constructed. The overall support system was completed when the platform is excavated, and settlement caused by the construction of the platform layer is 3.8%, which has little effect on the overall ground settlement during the whole excavation process.

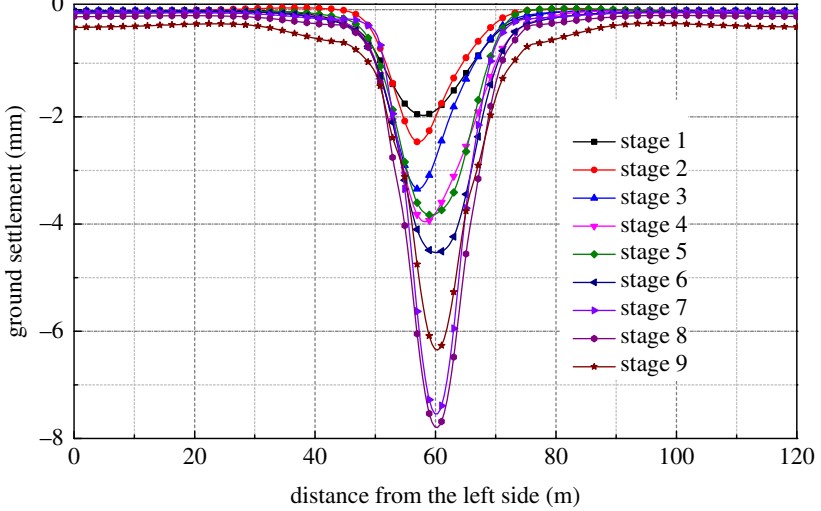

**Figure 4.** Surface settlement of the main construction sequences.

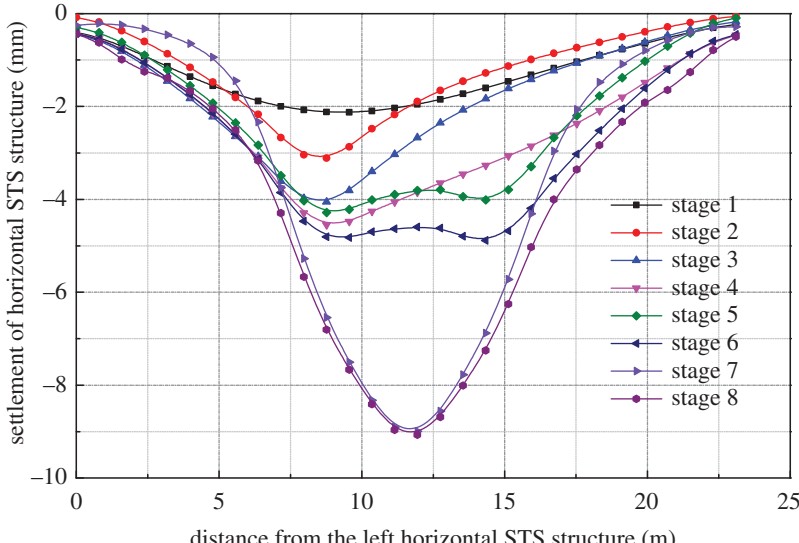

**Figure 5.** Vertical deformation of horizontal STS structure.

### 3.2.2. Vertical deformation of horizontal pipe-roofing structure

The vertical deformation curve of the horizontal STS structure is shown in figure 5. It can be seen from the figure that the construction of the bottom-left pilot tunnel is completed, followed by the horizontal STS structure reaching to a maximum, a maximum vertical deformation of 2.12 mm, and a maximum deformation point close to the left side. Further settlement occurs when the excavation of the bottom-left pilot tunnel is completed. Meanwhile, the horizontal STS structure also shows the deformation characteristics of the continuous beam. When the construction of the left column and longitudinal beam are completed, the structure of the left column and longitudinal beam is equivalent to a fixed fulcrum, which reduces the calculated length of the horizontal STS structure. Therefore, the effect of excavation of the right pilot tunnel on the total settlement is significantly reduced compared with that of the left pilot tunnel. The STS structure also presents obvious continuous beam characteristics during the excavation of the right-bottom pilot tunnel.

It can be seen from the figure that the construction of the mid-span buckle arch has the greatest influence on the settlement of the STS structure, and the construction of the mid-span increases the settlement of the STS structure while reducing the settlement of the side-span. The excavation of the station hall has little effect on the settlement of the mid-span, has a great influence on the left and right cross-span and has no significant impact on the overall settlement.

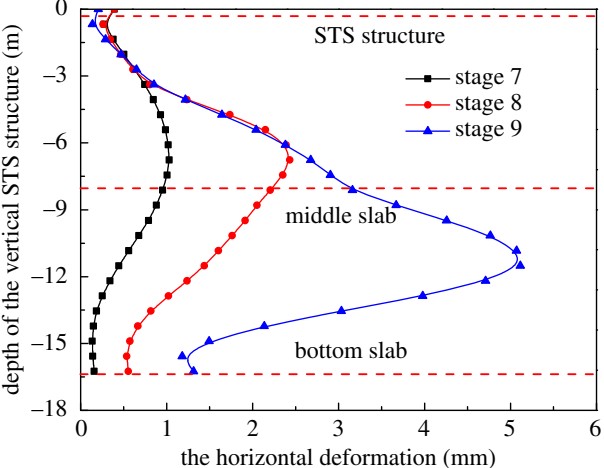

**Figure 6.** Horizontal deformation of vertical STS structure.

### 3.2.3. Horizontal deformation of vertical pipe-roofing structure

In the traditional PBA method, the side piles are generally used as the lateral supporting structure when the main structure is excavated, and the deformation of the lateral retaining structure directly affects the stability and safety of the entire structure. The STS method is applied as the lateral enclosure structure.

It can be seen from the numerical results that the deformation of the left STS structure is slightly larger. The horizontal deformation of the vertical STS structure on the left side is analysed, because it is necessary to study the stability and safety of the lateral support structure during the excavation period. From figure 6, after completion of the construction of the main body arching stage, the maximum horizontal displacement of the vertical STS structure is 1.03 mm, which is located near the middle plate. In the excavation stage of the station hall, the calculation of the length of the vertical STS structure is shortened due to the application of the temporary brace, which effectively limits the deformation of the STS structure, and the maximum horizontal displacement of the structure is 2.45 mm. According to the previous construction experience, the excavation of the platform side-span is the most dangerous construction stage of the entire main body excavation, and the vertical STS structure may suffer from strength failure and overall instability. It can be seen from figure 6 that the maximum horizontal displacement of the vertical STS structure is 5.14 mm, which meets the construction requirements.

## 3.3. Impact of construction sequence of pilot tunnel on ground settlement

Section 3.2 shows that the construction of the pilot tunnel and the mid-span buckle arch causes most of the total settlement. Combined with the construction experience, the excavation sequence of the pilot piles through the PBA method has a multi-cavern effect, and the stratum settlement can be significantly different using different excavation sequences. Therefore, it is of great significance to study the excavation sequence of the pilot tunnel in the STS method combined with the PBA method. The design schemes are shown in table 3.

The ground settlement curve after construction of each scheme is depicted in figure 7. It can be seen from the figure that the effect of the four schemes on the final settlement are almost the same. The settlement increases most rapidly in the excavation stage of the bottom pilot tunnel. The final settlements are 4.53 mm, 4.64 mm, 4.40 mm and 5.19 mm, respectively, and they are all on the top of the upper pilot tunnel. Compared with schemes 1 and scheme 2, the maximum settlement of the excavation of the bottom pilot tunnel is 0.11 mm which is higher than that of the lower pilot tunnel. This is because the excavation of the bottom pilot tunnel may lead to secondary disturbance to the stratum. In scheme 3, the advance construction of the top-right pilot tunnel causes the state of 'reverse arch' at the top of the left pilot tunnel, which reduces the settlement in the left pilot tunnel excavation. In scheme 4, due to the two continuous excavation of the bottom pilot tunnel, the disturbance to the stratum is large so the accumulated settlement is also the largest. For shallow tunnels, scheme 3 is the most reasonable construction sequence for excavation tunnels considering the ground settlement.

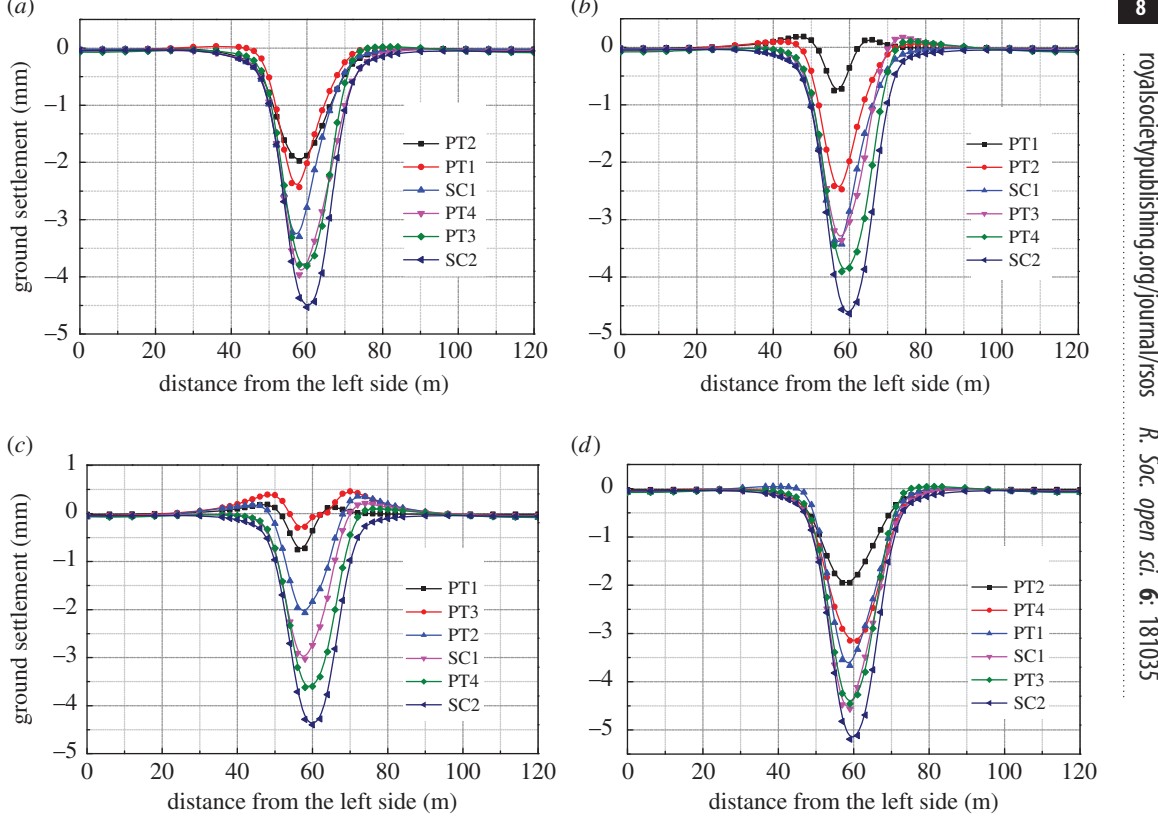

**Figure 7.** Surface settlement/ (*a*) scheme 1, (*b*) scheme 2, (*c*) scheme 3 and (*d*) scheme 4.

**Table 3.** Construction scheme.

| excavation sequences | Stage 1 | Stage 2 | Stage 3 | Stage 4 | Stage 5 | Stage 6 |
|---|---|---|---|---|---|---|
| scheme 1 | PT2 | PT1 | SC1 | PT4 | PT3 | SC2 |
| scheme 2 | PT1 | PT2 | SC1 | PT3 | PT4 | SC2 |
| scheme 1 | PT1 | PT3 | PT2 | SC1 | PT4 | SC2 |
| scheme 4 | PT2 | PT4 | PT1 | SC1 | PT3 | SC2 |

## 3.4. The influences of construction sequence of the station hall on deformation of ground and STS structure

### 3.4.1. Surface settlement

To study the influence of the construction sequences of the station hall on the ground settlement, the following two schemes are analysed: (i) scheme 1: first mid-span then side-span and (ii) scheme 2: first side-span then mid-span.

Based on the analysis of the excavation sequences of the two station halls, the maximum settlement of the ground surface is concentrated near the centre line of the station. The settlements of different construction stages are shown in table 4. It can be seen from the table that the maximum settlements caused by schemes 1 and 2 are 7.68 mm and 5.99 mm, respectively, and the settlement caused by plan 2 is reduced by 22% compared with plan 1. As the construction stage of the pilot tunnel and the middle column for the two schemes are the same, the settlement after the excavation of the station hall layer is analysed in figure 8. There will be little settlement after the completion of the construction of the station hall because the entire station's supporting system is basically completed. There is an uplift phenomenon during the main construction; therefore the maximum settlement takes place as a result of the station hall construction.

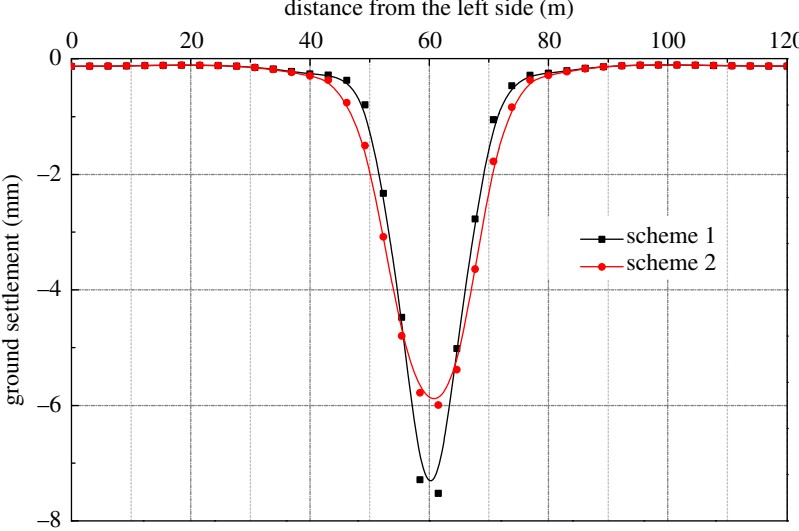

**Figure 8.** Surface settlement.

**Table 4.** Surface settlement of each construction stage.

| construction | PT2 | PT1 | SC1 | PT4 + PT3 + SC2 | platform |
|---|---|---|---|---|---|
| scheme 1 (mm) | 1.95 | 2.47 | 3.34 | 4.51 | 7.68 |
| scheme 2 (mm) | 1.95 | 2.47 | 3.34 | 4.51 | 5.99 |

According to the excavation sequence of scheme 2, numerical simulation is carried out for each construction stage, and the ground settlement curve is shown in figure 9. It can be seen from the figure that after the completion of the middle column construction, the maximum ground settlement is located near the centre line of the horizontal STS structure, approximately 4.48 mm. The STS structure deforms and the STS structure of the middle span uplifts when the left- and right-side span of the station hall are excavated. The maximum ground settlement is about 4.3 mm, which is located at the top part of the top-right pilot tunnel. After the construction of the mid-span and station halls, the final settlement is 6.0 mm.

In general, ground settlement increases slightly during the construction of the side-span while it has no major impact on the total deformation. During the excavation of the mid-span, the settlement increases by 1.64 mm and the excavation of the station hall has no impact on the total ground settlement.

### 3.4.2. Vertical deformation of STS structure

The vertical deformation of the STS structure caused by the excavation of the station hall in scheme 2 is shown in figure 10. After the excavation of the station hall, the vertical deformation of the side-span of the horizontal STS structure causes the mid-span uplift, about 1.02 mm, and the overall settlement of the STS structure presents a 'W' shape. When the mid-span excavation is completed, the vertical deformation of the STS structure reaches 6.35 mm. Compared with scheme 1, the vertical deformation of the STS structure reduced by about 2.5 mm. Therefore, scheme 2 is more conducive to controlling the vertical deformation of the STS structure in the station hall.

## 3.5. The effect of temporary supports on ground deformation and STS structure

### 3.5.1. The supporting scheme

In the excavation sequences of the station hall, a temporary steel support is set at the corner of the STS structure to reduce the lateral calculation length of the whole space and improve the bearing capacity and stability of the STS structure, which is shown in figure 11. To analyse the effects of temporary bracing on the ground settlement and the deformation of the STS structure, three schemes (without temporary

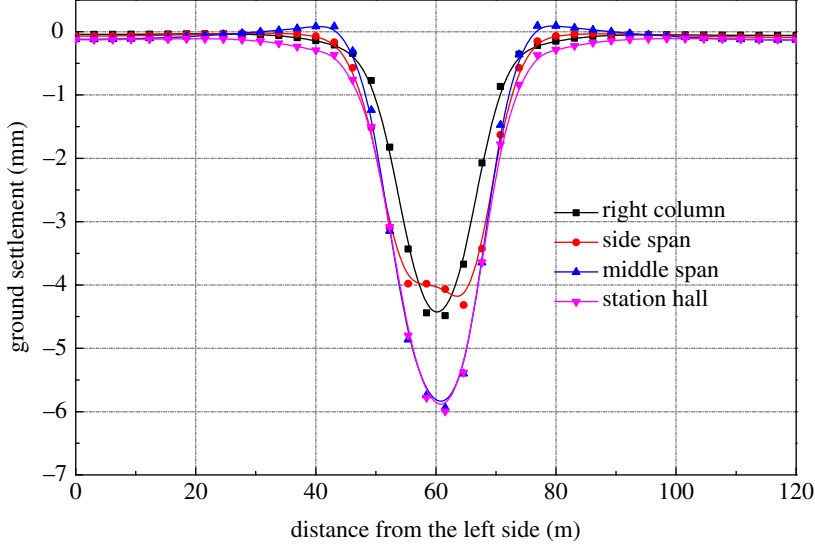

**Figure 9.** Surface settlement curve.

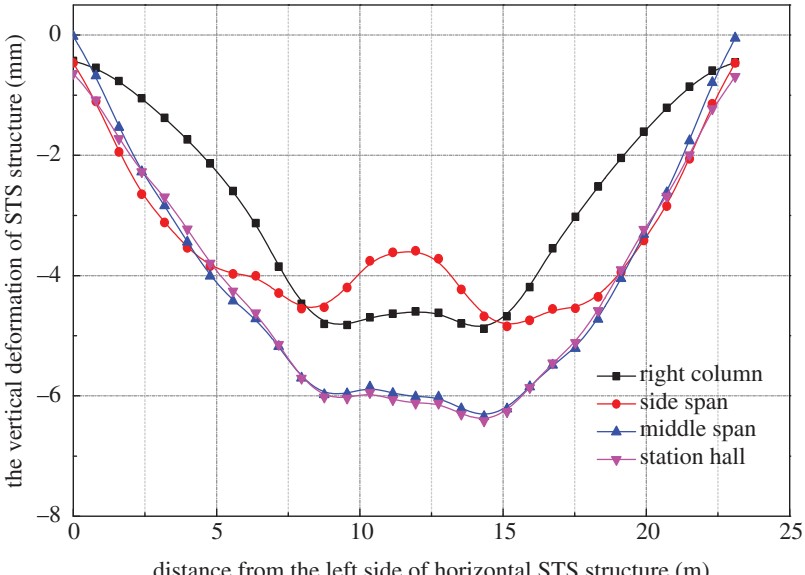

**Figure 10.** Vertical deformation of STS structure.

support, longitudinal spacing with 1 m and longitudinal spacing with 2 m, respectively) are investigated, which is shown in table 5.

### 3.5.2. The influences of temporary steel supports on surface settlement

The calculation span of the side-span can be shortened, and vertical load can be shared by the horizontal stiffness of the STS structure. As a result, the ground settlement and STS structure deformation can be significantly reduced by installing temporary steel supports. The ground settlement caused by each scheme is shown in figure 12. It can be seen from the figure that the maximum settlements caused by the three schemes are 8.68 mm, 7.80 mm and 7.70 mm, respectively. The results show that the temporary supports can effectively reduce the ground settlement. With the spacing of the temporary supports, the effect of the temporary supports on ground settlement will increase and then reach a constant.

### 3.5.3. The influences of temporary steel supports on vertical deformation of horizontal STS structure

It can be seen from figure 13 that the maximum vertical deformation of the horizontal STS structure caused by scheme 1 is 9.90 mm for the construction of the station hall, which is 0.84 mm greater than

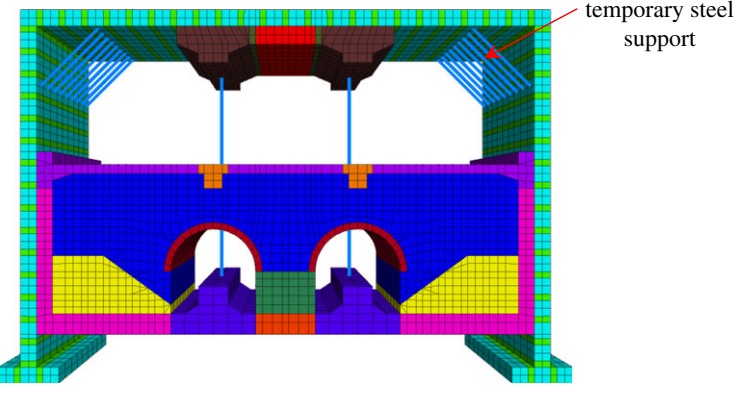

**Figure 11.** The temporary supports.

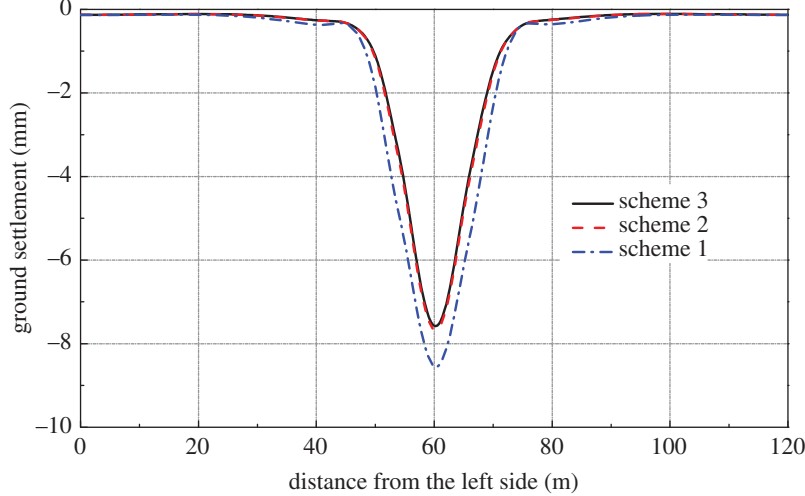

**Figure 12.** Surface settlement.

**Table 5.** Parameters of models.

| scheme | scheme 1 | scheme 2 | scheme 3 |
|---|---|---|---|
| Bracing span (m) | — | 2 | 1 |

that of scheme 2, and 0.90 mm larger than scheme 3. Therefore, the temporary supports directly affect the vertical deformation of the side-span parts because temporary supports shift a part of the vertical load to the sides of the STS structure. Moreover, it can be seen from the figure that the deformations of the STS structure of schemes 2 and 3 are basically the same.

### 3.5.4. The influences of temporary steel supports on horizontal deformation of vertical STS structure

It can be seen from figure 14 that the maximum horizontal deformation of the vertical STS structure caused by scheme 1 is 4.82 mm for the excavation of the station hall, which is located about 1 m above the middle plate and is increased by 2.45 mm compared with scheme 3. The deformation of schemes 2 and 3 are basically the same, and the maximum horizontal deformation difference is 0.18 mm. Therefore, the temporary supports have a great effect on the deformation of the horizontal STS structure and should be installed during the construction stage in order to ensure stability.

## 3.6. The effect of transverse stiffness of STS structure on surface settlement

Unlike the traditional pipe-curtain method, a lateral joint is installed in the STS structure which can effectively eliminate the need for complicated temporary supports and control the ground settlement.

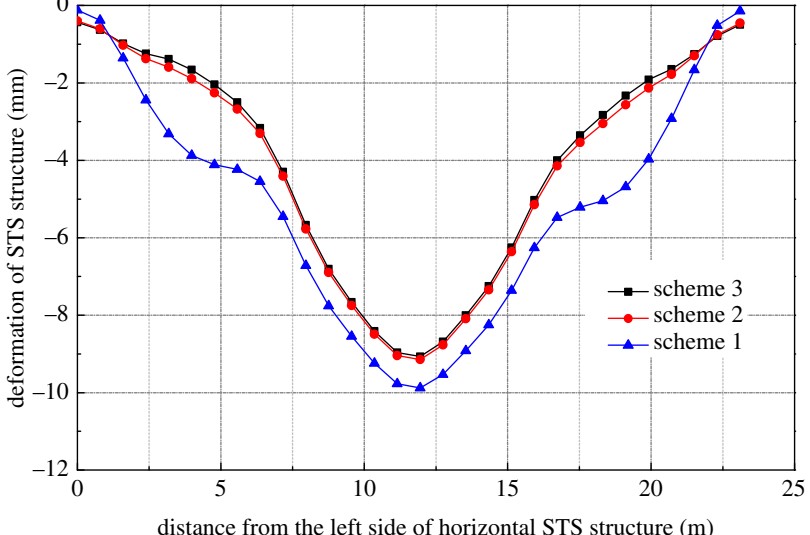

**Figure 13.** Vertical deformation of horizontal STS structure.

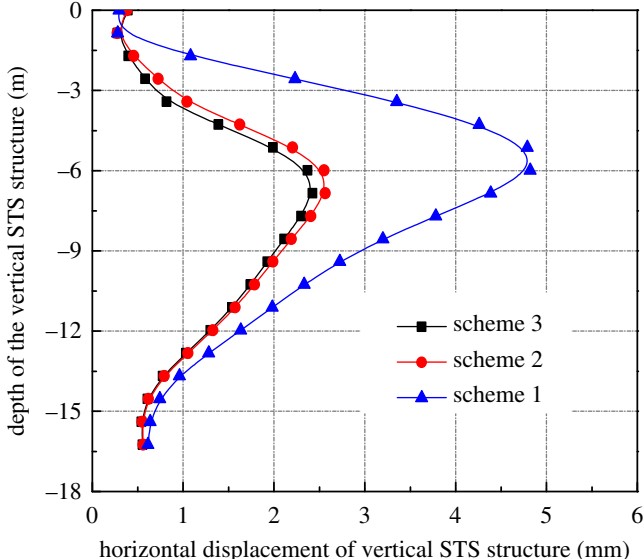

**Figure 14.** Horizontal deformation of vertical STS structure.

The lateral stiffness of the STS structure can directly reflect the deformation characteristics of the structure, and subsequently affects the deformation of the soil. Based on the laboratory experiment and numerical simulation, the load versus deformation curve is calculated and analysed. It can be concluded that the stiffness of the STS structure is $1.05 \times 10^8 \, \text{N} \cdot \text{m}^2$ when the flange plate is un-welded, and the stiffness is $1.91 \times 10^8 \, \text{N} \cdot \text{m}^2$ when the flange plate is welded.

To simplify the lateral stiffness of the STS structure, the lateral stiffness is the same as concrete. The calculation formula of the rectangular section stiffness is

$$EI = E\frac{bh^3}{12}, \tag{3.1}$$

where, $b$ is 0.5 m, the design strength of concrete is C30, and the elastic modulus $E$ is reduced to 0.8 times, about 24 GPa. The calculation result is shown in table 6.

In the field construction process of the STS structure, the influence of the pore-forming quality, the bolt installation and the flange plate welding on the stiffness of the STS structure means the lateral stiffness is not as perfect as in laboratory conditions. In the calculation process, it is necessary to reduce the lateral stiffness to different degrees according to the quality of the construction. According

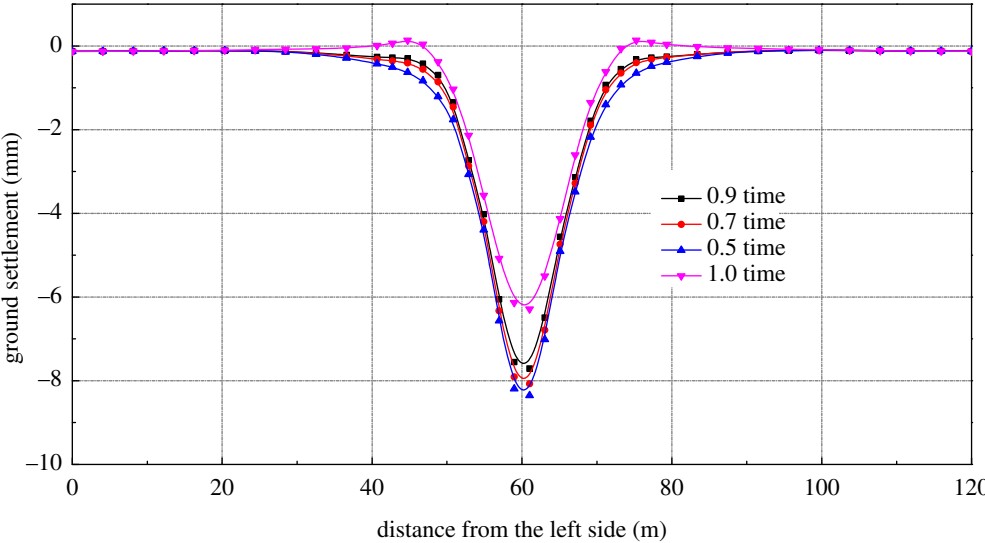

**Figure 15.** The surface settlement in the different stiffness of STS structure.

**Table 6.** The size of the equivalent structure.

| stiffness reduction | 1 time | 0.9 time | 0.7 time | 0.5 time |
|---|---|---|---|---|
| no-welded flange thickness (mm) | 472 | 455 | 419 | 374 |
| welded flange thickness (mm) | 576 | 556 | 511 | 457 |

**Table 7.** Simplified calculation parameters of STS structure. $E_n$ is the elastic modulus when the flange is welded; $E_y$ is the elastic modulus when the flange is not welded.

| stiffness reduction | Material 1 $E$ (GPa) | Material 2 $E_n$ (GPa) | Material 2 $E_y$ (GPa) | $\mu$ |
|---|---|---|---|---|
| 0.9 | 65 | 1.58 | 3.00 | 0.25 |
| 0.7 | 65 | 1.21 | 2.30 | 0.25 |
| 0.5 | 65 | 0.85 | 1.59 | 0.25 |

to the simplified results of the STS structure provided by Jia [6], the stiffness of the STS structure is reduced by 0.5 times. The specific parameters are shown in table 7.

In terms of ultra-shallow-buried tunnels and underground stations, the settlement of the vault caused by excavation is predicted by the ground settlement, which means ground settlement control is one of the important factors during construction. The ground settlement for different STS structure stiffness is shown in figure 15. From the figure, the stiffness of the STS structure can reduce the ground settlement significantly when reduced by 0.5 times.

# 4. Description of engineering project in China

Subway stations in Shenyang (line 10) are constructed by the traditional PBA method that has some challenges when used in Northeastern Street station (located west of the intersection of Beihai street and Northeastern Road) due to the complexity of the existing structures and heavy traffic load. As a result, considering the geographical location and geological properties of the subway station, a combination of STS and PBA methods, as discussed in the previous section, is used to design and construct this station.

The total length of the station is 225.95 m, the width of a standard cross section is 24.7 m, and the buried depth of the bottom floor of the station is 17.5 m, with an overburden layer of about 3.4 m. For

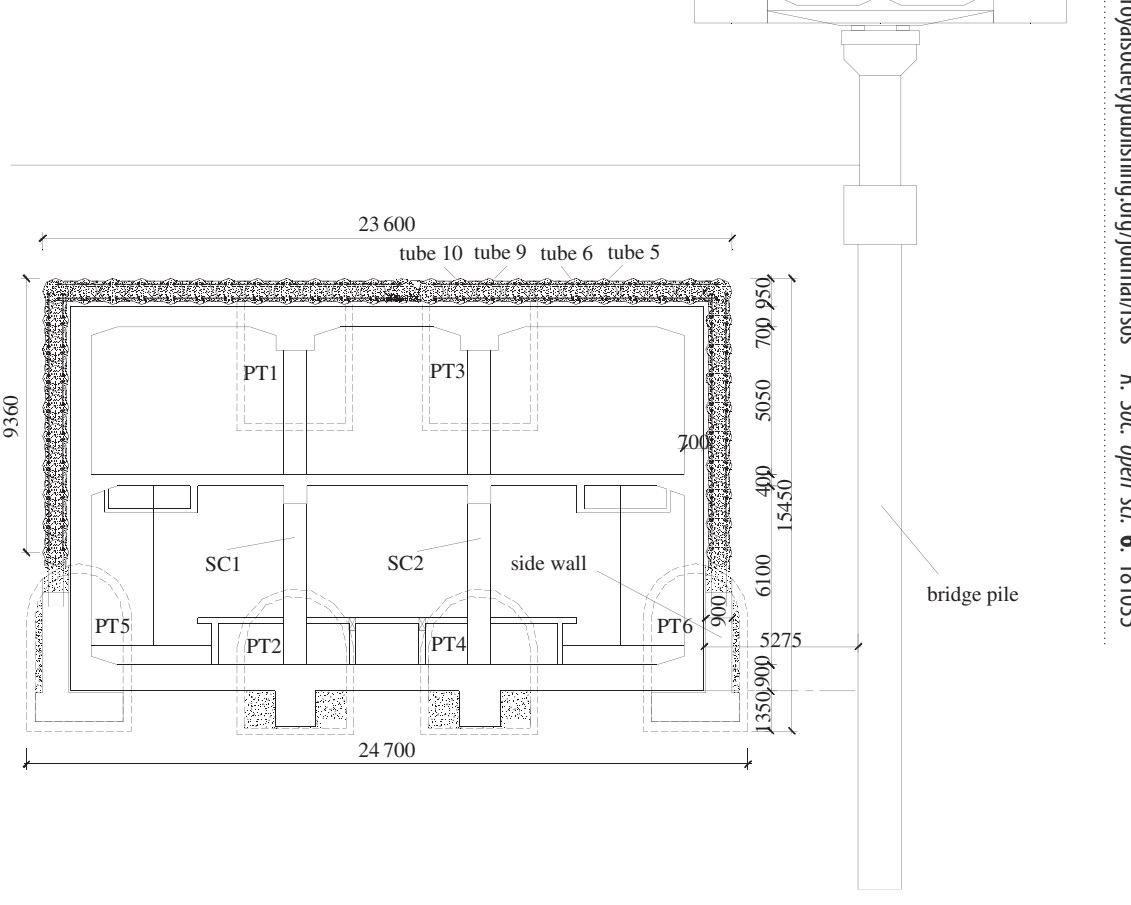

**Figure 16.** Cross section of the Northeastern Street Station.

construction, the station is divided into three sections: north of the station for which covered excavation is used; south of the station for which the open surface method is adopted and the middle section of the station in which a combination of the STS and PBA methods is adopted considering the intensity of the surrounding structures and heavy surface traffic load. The length of the southern, middle and the northern segments of the station are 68.65 m, 114.4 m, 42.9 m, respectively. The engineering details of the Northeastern Street station are illustrated in figure 16.

The soil stratum at the station location was determined from geological investigations. Accordingly, the field can be classified into five layers based on the mechanical properties and formation lithology, as shown in table 8. And the soil profile was shown in figure 17 [6].

## 5. *In situ* monitoring

The bolt stress is changed by the deformation of the STS structure during the construction process. In this research, the stress gauges were installed at the bolt location to monitor the stress change. According to the site condition, the monitoring points between 5# and 6# steel tubes and between 9# and 10# steel tubes are selected, which was shown in figure 16. Eight monitoring points were arranged along the axial direction of the steel tube with spacing of about 5 m (figure 18). The monitoring instruments were JTM-V1000 vibrating strings and a GPC-3-type digital auto-reading instrument.

Research results in chapter 3 shows the construction of the pilot tunnel had a great influence on the deformation of the STS structure and the ground settlement. Therefore, the influence of construction of pilot tunnel on the changing rule of bolt stress is investigated. Two stages of construction sequences are analysed under the site construction condition: tunnel excavation and beam-column system construction. Let's take the case of the steel tubes 5#-6#.

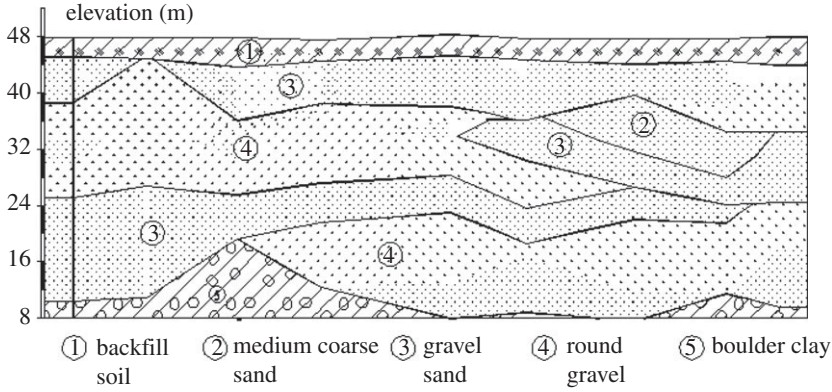

**Figure 17.** Geological profile of the soil.

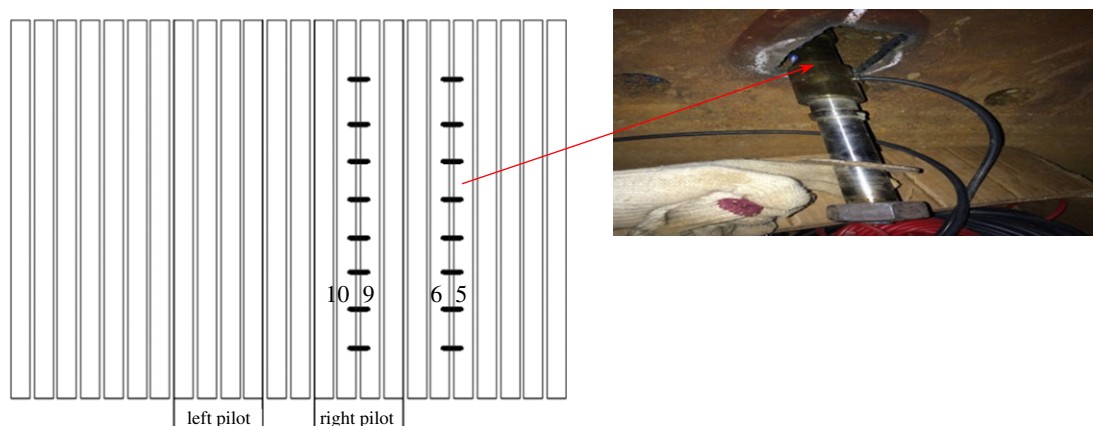

**Figure 18.** Arrangement of monitoring sensors.

**Table 8.** Soil stratigraphy and properties at the site.

| soil | thickness (m) | internal friction angle (°) | cohesion (kPa) | Poisson ratio | elasticity modulus (MPa) | density (kg m$^{-3}$) |
|---|---|---|---|---|---|---|
| backfill soil | 3.4 | 24 | 3 | 0.28 | 16 | 1800 |
| gravel sand | 6 | 33 | 2 | 0.26 | 22.4 | 2000 |
| round gravel | 11 | 34 | 2 | 0.25 | 27 | 2050 |
| medium coarse sand | 10 | 34 | 2 | 0.29 | 19 | 1980 |
| boulder clay | non-penetrating layer | 27.5 | 7 | 0.30 | 23 | 2000 |

(1) Construction of pilot tunnel: It can be seen from figure 19 that when the concrete retaining piles of the outside pilot tunnel were broken, the stress of the bolt started to change. The stress of the bottom stress gauges increased rapidly with the excavation. The tensile force changed from 0.7 kN to 6.32 kN, and the variation was 7.02 kN. When the bottom-right pilot tunnel was excavated, the steel bolts were in the pressure state, indicating that the lateral overall performance of the STS structure was good. The bottom flange plates were welded to enhance the lateral stiffness and stability of the STS structure and overcome the weak connection of the traditional pipe-roofing structure. After the excavation of the pilot tunnels, the stress value of the gauges still increased slightly. It showed that there was a certain time effect when the overburden soil was excavated and unloaded.

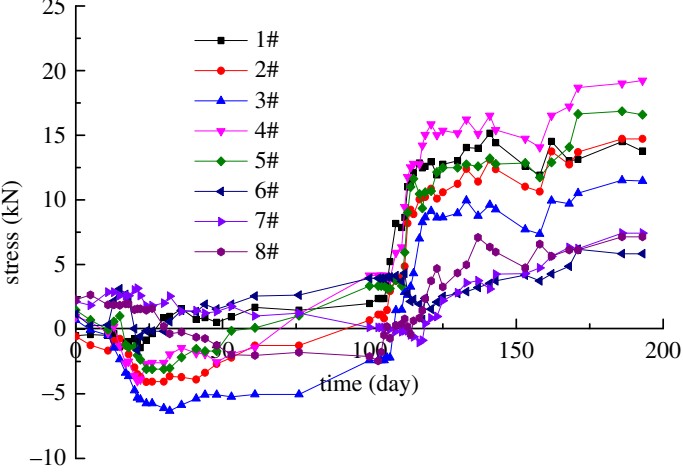

**Figure 19.** The stress value along with excavation sequences.

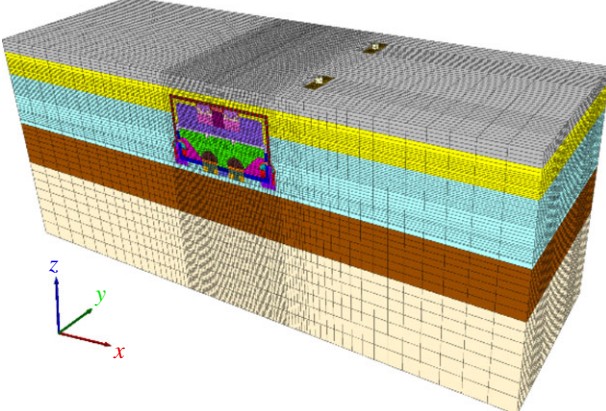

**Figure 20.** Three-dimensional difference model of the station construction.

(2) Construction of the beam-column system: at this stage, the value was basically maintained in a stable state by monitoring the stress gauges, but the overall trend was increasing, and the data reached a constant, which was related to the multiple disturbance of the soil during the construction of the pilot tunnel.

# 6. Numerical modelling of engineering application

## 6.1. Model and material properties

A large span underground subway station is treated as a three-dimensional model. A full three-dimensional model of the station construction was established using FLAC$^{3D}$ software (FLAC$^{3D}$ 5.0). The dimensions of the numerical model were $120 \times 39 \times 50$ m to eliminate the boundary conditions [6], as shown in figure 20. The bottom of the model was fixed in $X$-, $Y$- and $Z$-directions. The front and back side were fixed in $X$- and $Y$-directions. The left and right side were fixed in $X$-and $Y$-directions. The top surface was free to move in all directions. A surcharge of 20 kPa was applied on the top surface to simulate surface loads. The diameter of the cylindrical steel columns is 0.8 m, the distance between cylindrical steel columns is 8.5 m, the length of the feet-lock bolt is 2.5 m and the distance between the feet-lock bolts is 1.0 m in the model, in accordance with the real structure.

Beam and pile elements are adopted to simulate the temporary steel supports and feet-lock bolt, respectively. Different materials were assigned to the different soil layers and different components of the STS structure as shown in figure 20. Line and shell elements were used to simulate the cylindrical steel column and the primary lining, respectively. For simplicity, the Mohr–Coulomb model was adopted to define the behaviour of the natural and improved soil in the numerical analyses. STS

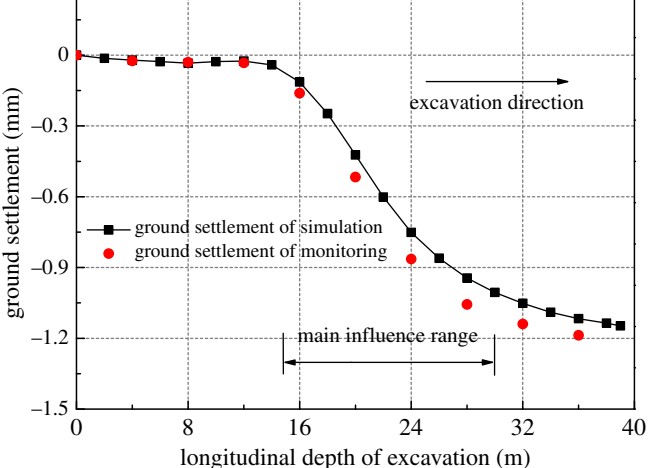

**Figure 21.** Crown settlement profile for tunnel PT1.

structure elements, primary lining, beam and columns are deemed to have a liner-elastic behaviour. Material properties used for the numerical simulations are listed in tables 1 and 2.

In accordance with the construction sequence of Northeastern Road subway station, the whole construction simulation was divided into six stages, as follows:

(1) Stage 1: Construction of the side pilot tunnels PT5 and PT6;
(2) Stage 2: Construction of the pilot tunnels PT1 and PT2 and cylindrical steel column SC1;
(3) Stage 3: Construction of the pilot tunnels PT3 and PT4 and cylindrical steel column SC2;
(4) Stage 4: Construction of the top plate at mid-span;
(5) Stage 5: Construction of the station hall; and
(6) Stage 6: Construction of the platform layer.

## 6.2. Verification of the numerical model

The longitudinal displacement profile of the pilot tunnel Z1 was investigated. The step length was assumed to be 3 m in the numerical simulations in accordance with the realities of construction.

As for the ground settlement of the pilot tunnel, the simulation and monitored results showed similar settlement profiles when pilot tunnel PT1 was excavated, and the largest difference of both was within 10%, as shown in figure 21. The numerical model was reasonable and correct, which can be used for further analysis.

## 6.3. The influence of construction sequences on bridge piles

The Dongbei Damalu Station is adjacent to the viaduct near Beihai Street. The horizontal distance between the viaduct and subway station is only about 4 m. Therefore, it is necessary to strictly control the settlement and horizontal displacement of the viaduct during the construction of the subway station. The photo of the viaduct is shown in figure 22.

### 6.3.1. Settlement of bridge piles

During the buried excavation process of the subway station, the settlement of the bridge piles cap is shown in figure 23. In the stage of mid-span construction, the settlement cloud diagram of the piles is shown in figure 24. It can be seen from the figures that the settlement of the bridge-pile increases with time. The settlement of the 1# pile near the launching shaft was the largest, approximately 1.2 mm. The settlement of the bridge-pile is mainly caused by the horizontal deformation of the pile, and the top of the pile is the largest. The settlement of the bridge-pile is mainly concentrated in the pilot tunnel stage, the column and the second construction of mid-span.

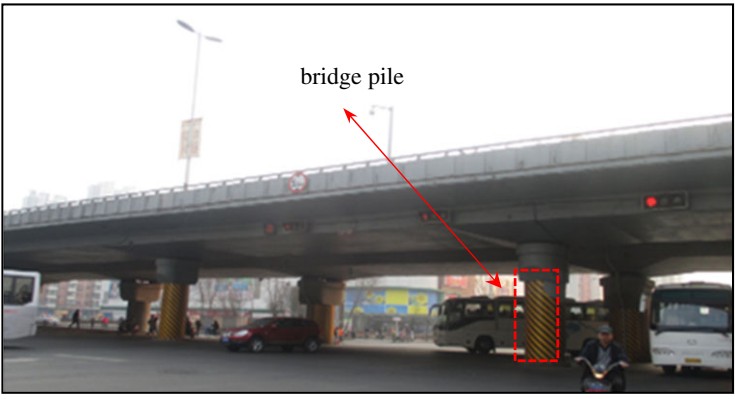

**Figure 22.** The adjacent viaduct.

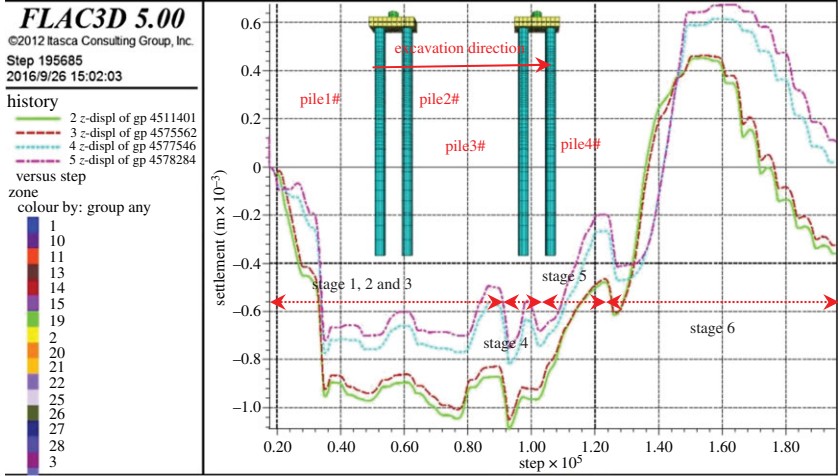

**Figure 23.** The settlement curve in the top of the bridge-pile.

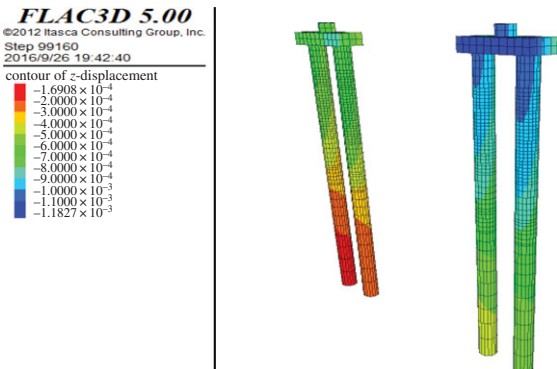

**Figure 24.** The settlement cloud image of bridge-piles.

### 6.3.2. The horizontal deformation of bridge piles

The horizontal deformation of the bridge pile after the main body construction is shown in figure 25. It can be seen from the figure that the deformation of each bridge pile is basically the same, and it was not sensitive to the excavation time compared to the ground settlement. The horizontal displacement of the top of the pile was the largest. The maximum horizontal displacement of the top of pile 1# was 2.80 mm, and the maximum horizontal displacement of the top of pile 4# was 2.64 mm. The horizontal displacement of the bottom of pile 4# was the largest, and the maximum displacement was about 0.80 mm. The main body construction had a great influence on the horizontal displacement in the middle part of the bridge pile.

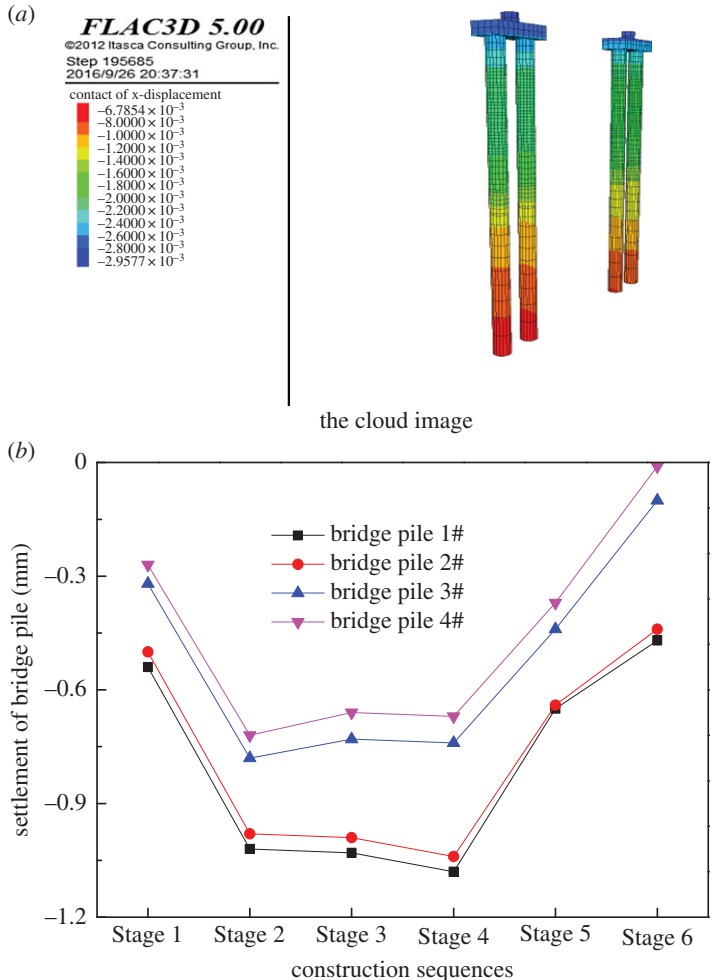

**Figure 25.** The horizontal deformation of bridge piles after main construction. (*a*) The cloud image and (*b*) the settlement curve.

## 6.4. The influence of side pilot tunnel on bridge piles

### 6.4.1. The supporting scheme of right-side pilot tunnels

To analyse the influence of concrete grouting on the side of the pilot tunnel and the spacing of temporary support on the deformation of the bridge pile, the following four schemes were used for the simulation calculation, and the characteristic monitoring points were shown in figure 26.

- Scheme 1: The side of the pilot tunnel was grouted to reinforce soil, and the spacing of steel supports was 0.5 m.
- Scheme 2: The side of the pilot tunnel was grouted to reinforce soil, and the spacing of steel supports was 2.0 m.
- Scheme 3: The side of the pilot tunnel was grouted to reinforce soil, and steel supports had not been installed.
- Scheme 4: There is no need to reinforce soil and install the steel supports in the construction of the pilot tunnel.

### 6.4.2. Settlement of right-side pilot tunnel

It can be seen from figure 27 that the settlement at point A has an obvious time effect. The first three schemes had little effect on the settlement of point A. In scheme 4, no reinforcement support was used, and the ground settlement of point A increased about 1 mm. The horizontal displacement of the pile side of the right pilot tunnel was quite different in four schemes with scheme 4 taken as an example. Within a certain influence range of the bridge pile, the horizontal displacement of point B is small. It may be that the bridge pile was similar to the support pile and bore the horizontal earth pressure.

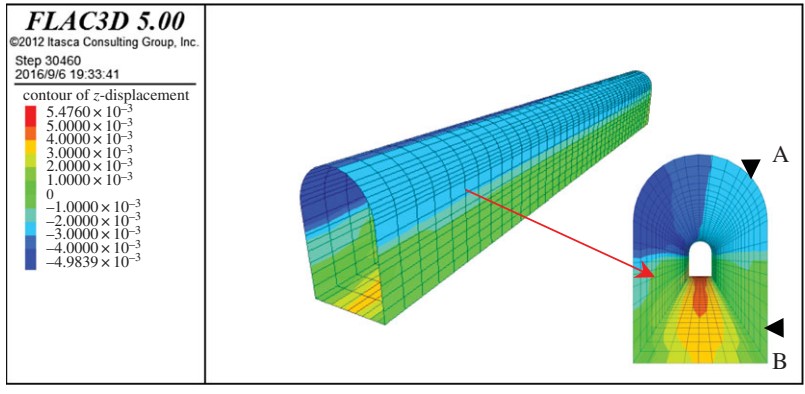

**Figure 26.** The monitoring points.

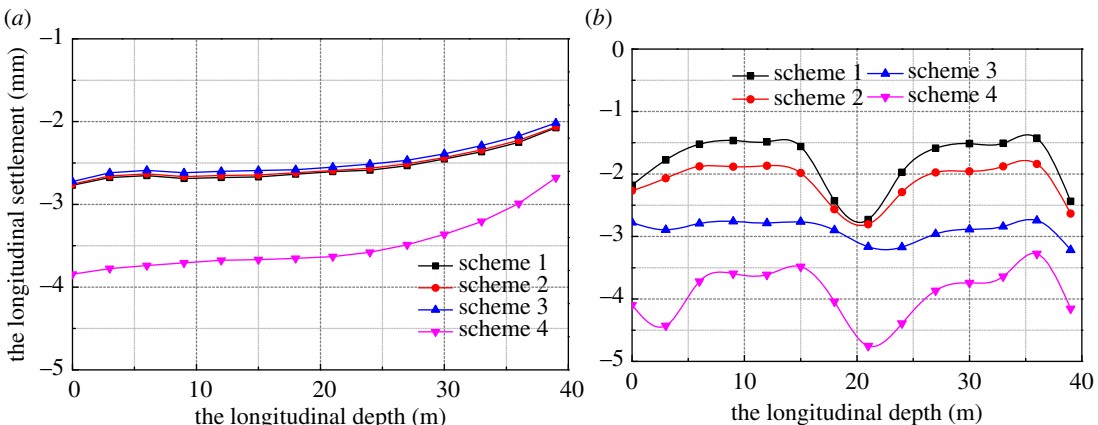

**Figure 27.** The horizontal displacement. (*a*) A point and (*b*) B point.

### 6.4.3. Deformation of bridge piles

The maximum horizontal displacement of four schemes is shown in figure 28. The calculation results of schemes 1–3 are very similar, but scheme 1 of the strong support has the largest horizontal displacement and settlement. Pile #1 serves as an example to analyse the influence of the different supporting scheme on the horizontal displacement of the bridge pile. The pile can effectively constrain the ground settlement, and the horizontal displacement of the pile top is the largest, as during the excavation of the pilot tunnels, the influence range of the tunnel excavation was small in the sand layer. However, in scheme 4, the horizontal displacement of the pile top was the largest. Scheme 3 was the optimum choice compared with the other supporting schemes.

## 7. Conclusion

This study introduces the STS method that has been applied for the first time to construct an ultra-shallow-buried and large span subway station. Based on the numerical simulation and monitoring data, surface settlement and structural deformation in a large span subway station using the Steel Tube Slab were investigated. As a result, conclusions were drawn as follows:

(1) The STS structure can effectively control the deformation of ground and adjacent structures by connecting the steel pipes in a transversal direction and can be widely used in large span underground engineering in soft soil areas.

(2) The STS method and PBA method are combined to construct the subway station in Shenyang. The ground settlement is mainly accumulated during the construction of pilot tunnels and the secondary lining of mid-span. The construction of the main structure has little influence on the ground settlement, while it can cause horizontal deformation of STS structures.

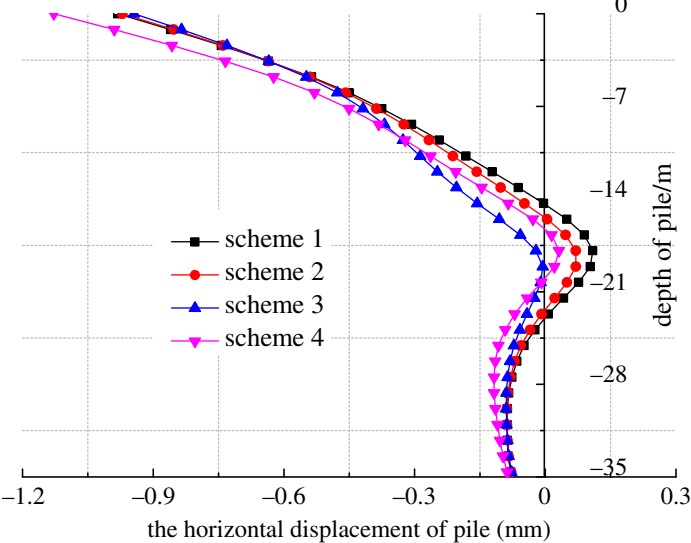

**Figure 28.** Horizontal displacement of 1# pile.

(3) As for construction of the station hall, the ground settlement of different construction sequences, i.e. rear-span to mid-span and mid-span to rear-span, can be reduced by 22%. Temporary supports can effectively reduce the deformation of the vertical STS structure, as well as controlling the deformation of the horizontal STS structure. The optimum spacing of temporary supports is 2 m.

(4) The side pilot tunnel has a significant effect on the deformation of the bridge pile, and the surrounding soil of pilot tunnel needs to be reinforced during the construction process.

Data accessibility. The datasets supporting this article have been uploaded as part of the electronic supplementary material.

Authors' contributions. Data collection: C.Z.Z. and P.J.J.; formal analysis: P.J.J. and C.Z.Z.; investigation: P.J.J., Z.G.W. and Q.B.; writing—original draft: P.J.J. and X.D.; writing—review and editing: W.Z. and Y.C.

Competing interests. The authors declare no competing financial interests.

Funding. The research described in this paper was supported by the Fundamental Research Funds for the Central Universities (grant no. N160106006), the National Natural Science Foundation of China (grant nos. 51578116 and 51878127) and China Scholarship Council.

Acknowledgement. These supporters are gratefully acknowledged. And the data are available for the journal.

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
