## [Reviewer comments · Royal Society Open Science]

Review History

RSOS-181035.R0 (Original submission)

Review form: Reviewer 1

Is the manuscript scientifically sound in its present form?

Yes

Are the interpretations and conclusions justified by the results?

Yes

Is the language acceptable?

Yes

Is it clear how to access all supporting data?

Yes

Do you have any ethical concerns with this paper?

No

Have you any concerns about statistical analyses in this paper?

No

Recommendation?

Accept with minor revision (please list in comments)

Comments to the Author(s)

This paper describes an interesting pre-supporting system, called Steel Tube Slab (STS), that can be useful especially in case of large span underground excavation close to ground surface. Based on the numerical simulation and in situ monitoring, STS structure seems to be very effective and dependable measure in controlling and reducing the surface settlement and the existing surrounding buildings. Moreover, an interesting section of the paper is related to the influence of the excavation step length on the horizontal displacements at the tunnel face and on the ground settlement. I think that it's very useful for the application and promotion of STS technique. However, In order to improve the level of this manuscript better, some revisions need to be done, such as the following.

1. "I suggest the Authors highlighting/mentioning the main findings obtained at the end of the introduction section. "Several key parameters in the design ..." is too generic."
2. The problem studied by the Authors is a very complex soil-structure interaction problem, and I have appreciated the workflow of the paper, nevertheless I think that the soil conditions at the site are in general poorly described. So, if it is possible I recommend providing 1 Figure.
3. Section 5.2. I clearly understand the complexity of such a numerical model, however why did the Authors select the Mohr-Coulomb model? It can be considered sufficient for this case-study?
4. Fig. 20 - Why are used 0.7 m long element in simplified model of STS structure.
5. Fig.24. the numerical solutions presented in Fig. 24 are direct predictions or the result of a back-analysis.
6. Page 2, Line 12. "in-suit" should be "in-situ...". This error is repeated several times in all the paper;
7. Page 5, Line 54. "stain" should be "station";
8. Page 8, Line 15. Please check the value of 0.022mm, I think it is not correct. Probably should be 16 mm.
9. Page 15, Lines 31-32. "It indicate that ... were validity" I think it should be better to change with "It indicates that ... were valid";
10. Page 16, Line 29. "of STS structure has s high stiffness" should be "of STS structure has a high stiffness";

Review form: Reviewer 2 (Baofeng Jiang)

Is the manuscript scientifically sound in its present form?

Yes

Are the interpretations and conclusions justified by the results?

Yes

Is the language acceptable?

Yes

Is it clear how to access all supporting data?

Yes

Do you have any ethical concerns with this paper?

No

Have you any concerns about statistical analyses in this paper?

No

Recommendation?

Accept as is

Comments to the Author(s)

1. The title is too broad, a new title "Study on the effect of combining the new supporting structure with PBA method for a large span subway underground station" is suggested.
2. The relative literatures can be added in the introduction.
3. Some spelling and grammatical errors need to be modified, such as page2 "in-suit", page5 "statin", etc. Check all the paper.
4. Page 11: Usually the Puason coefficient is indicated by " ν ".
5. In Fig.17, it is not clear what does the colored part represent.
6. The length of the paper is a little long, and the length should be reduced properly.

Review form: Reviewer 3

Is the manuscript scientifically sound in its present form?

Yes

Are the interpretations and conclusions justified by the results?

Yes

Is the language acceptable?

Yes

Is it clear how to access all supporting data?

Yes

Do you have any ethical concerns with this paper?

No

Have you any concerns about statistical analyses in this paper?

No

Recommendation?

Accept with minor revision (please list in comments)

Comments to the Author(s)

Brief summary

STS construction method is first to be proposed, and according to the numerical calculation and the field monitoring, the authors obtain that STS structure can reduce and control settlements and reduce unpleasant effects on existing buildings in the neighborhoods of the excavation site. It is a

good paper for the later engineering application, but I have some concerns/doubts about the manuscript and suggest the author make several minor versions.

Comments and Questions

- 1) Fig. 3 (left) is similar to Fig. 6 (f). on the other hand, Fig. 3 (right) is identical to Fig. 7. Please check that.
 - 2) Table 2. How to get submitted E?
 - 3) As for the numerical model, the relationship between soil and structure is a complex question, how did the Authors consider?
 - 4) In section 5.3, there should have the construction history depending on the staged construction. The construction history will justify the heave, which is observed in Fig. 24.
 - 5) From the framework of this paper, it seems that the fourth chapter is not strictly necessary, I suggest the author delete the part to increase the continuity of this paper. If it is possible, the author can make some the initial parameter analysis in this chapter using the numerical software.
- Typo and grammatical
- 1) In suit? It should be "in situ", for example Page 2 Row 11, please check the text.
 - 2) "...a new technique for are presented..." should be "...a new technique for ... is presented" in the Page 5, Row 7.
 - 3) "it was decided ... , is used to design and construct the station" I think is better to change with "it was decided to use ... for the design and the construction of the station " in the Page 5, Rows 47-48
 - 4) The cross-references to Fig. 9a,b,c,d,e are not correct. I think these references are related to Fig. 6 in the Page 7.
 - 5) "he overburden pressure" should be "the overburden pressure" in the Page 8, Row 3.
 - 6) Page 16, Row 29: of STS structure has s high stiffness? Maybe it should be "a".
 - 7) "flange flanges" in the Page 23 Row 3?

Decision letter (RSOS-181035.R0)

26-Sep-2018

Dear Dr Jia,

The editors assigned to your paper ("A Case Study on the Application of the Steel Tube Slab Structure in Construction of a Subway Station") have now received comments from reviewers. We would like you to revise your paper in accordance with the referee and Associate Editor suggestions which can be found below (not including confidential reports to the Editor). Please note this decision does not guarantee eventual acceptance.

Please submit a copy of your revised paper before 19-Oct-2018. Please note that the revision deadline will expire at 00.00am on this date. If we do not hear from you within this time then it will be assumed that the paper has been withdrawn. In exceptional circumstances, extensions may be possible if agreed with the Editorial Office in advance. We do not allow multiple rounds of revision so we urge you to make every effort to fully address all of the comments at this stage. If deemed necessary by the Editors, your manuscript will be sent back to one or more of the original reviewers for assessment. If the original reviewers are not available, we may invite new reviewers.

To revise your manuscript, log into <http://mc.manuscriptcentral.com/rsos> and enter your Author Centre, where you will find your manuscript title listed under "Manuscripts with Decisions." Under "Actions," click on "Create a Revision." Your manuscript number has been

appended to denote a revision. Revise your manuscript and upload a new version through your Author Centre.

- Data accessibility

If you wish to submit your supporting data or code to Dryad (<http://datadryad.org/>), or modify your current submission to dryad, please use the following link:
<http://datadryad.org/submit?journalID=RSOS&manu=RSOS-181035>

- Competing interests

- Authors' contributions

- Acknowledgements

- Funding statement

Please note that Royal Society Open Science charge article processing charges for all new submissions that are accepted for publication. Charges will also apply to papers transferred to Royal Society Open Science from other Royal Society Publishing journals, as well as papers submitted as part of our collaboration with the Royal Society of Chemistry (<http://rsos.royalsocietypublishing.org/chemistry>). If your manuscript is newly submitted and subsequently accepted for publication, you will be asked to pay the article processing charge, unless you request a waiver and this is approved by Royal Society Publishing. You can find out more about the charges at <http://rsos.royalsocietypublishing.org/page/charges>. Should you have any queries, please contact openscience@royalsociety.org.

on behalf of Prof. R. Kerry Rowe (Subject Editor)
openscience@royalsociety.org

Associate Editor's comments:

Please ensure that you not only address the scientific concerns of the referees but also seek advice from a suitable language polishing service (see <https://royalsociety.org/journals/authors/language-polishing/> for instance) for advice to resolve the linguistic issues identified by the referees.

Comments to Author:

Reviewers' Comments to Author:

Reviewer: 1

Comments to the Author(s)

This paper describes an interesting pre-supporting system, called Steel Tube Slab (STS), that can be useful especially in case of large span underground excavation close to ground surface. Based on the numerical simulation and in situ monitoring, STS structure seems to be very effective and dependable measure in controlling and reducing the surface settlement and the existing surrounding buildings. Moreover, an interesting section of the paper is related to the influence of the excavation step length on the horizontal displacements at the tunnel face and on the ground settlement. I think that It's very useful for the application and promotion of STS technique. However, In order to improve the level of this manuscript better, some revisions need to be done, such as the following.

1. "I suggest the Authors highlighting/mentioning the main findings obtained at the end of the introduction section. "Several key parameters in the design ..." is too generic."
2. The problem studied by the Authors is a very complex soil-structure interaction problem, and I have appreciated the workflow of the paper, nevertheless I think that the soil conditions at the site are in general poorly described. So, if it is possible I recommend providing 1 Figure.
3. Section 5.2. I clearly understand the complexity of such a numerical model, however why did the Authors select the Mohr-Coulomb model? It can be considered sufficient for this case-study?
4. Fig. 20 - Why are used 0.7 m long element in simplified model of STS structure.
5. Fig.24. the numerical solutions presented in Fig. 24 are direct predictions or the result of a back-analysis.
6. Page 2, Line 12. "in-suit" should be "in-situ...". This error is repeated several times in all the paper;
7. Page 5, Line 54. "statin" should be "station";
8. Page 8, Line 15. Please check the value of 0.022mm, I think it is not correct. Probably should be 16 mm.
9. Page 15, Lines 31-32. "It indicate that ... were validity" I think it should be better to change with "It indicates that ... were valid";
10. Page 16, Line 29. "of STS structure has s high stiffness" should be "of STS structure has a high stiffness";

Reviewer: 2

Comments to the Author(s)

1. The title is too broad, a new title "Study on the effect of combining the new supporting structure with PBA method for a large span subway underground station" is be suggested.
2. The relative literatures can be added in the introduction.
3. Some spelling and grammatical errors need to be modified, such as page2 "in-suit", page5 "statin", etc. Check all the paper.
4. Page 11: Usually the Puason coefficient is indicated by " ν ".
5. In Fig.17, it is not clear what does the colored part represent.
6. The length of the paper is a little long, and the length should be reduced properly.

Reviewer: 3

Comments to the Author(s)

Brief summary

STS construction method is first to be proposed, and according to the numerical calculation and the field monitoring, the authors obtain that STS structure can reduce and control settlements and reduce unpleasant effects on existing buildings in the neighborhoods of the excavation site. It is a good paper for the later engineering application, but I have some concerns/doubts about the manuscript and suggest the author make several minor versions.

Comments and Questions

- 1) Fig. 3 (left) is similar to Fig. 6 (f). on the other hand, Fig. 3 (right) is identical to Fig. 7. Please check that.
- 2) Table 2. How to get submitted E?
- 3) As for the numerical model, the relationship between soil and structure is a complex question, how did the Authors consider?
- 4) In section 5.3, there should have the construction history depending on the staged construction. The construction history will justify the heave, which is observed in Fig. 24.
- 5) From the framework of this paper, it seems that the fourth chapter is not strictly necessary, I suggest the author delete the part to increase the continuity of this paper. If it is possible, the author can make some the initial parameter analysis in this chapter using the numerical software.

Typo and grammatical

- 1) In suit? It should be "in situ", for example Page 2 Row 11, please check the text.
- 2) "...a new technique for are presented..." should be "...a new technique for ... is presented" in the Page 5, Row 7.
- 3) "it was decided ... , is used to design and construct the station" I think is better to change with "it was decided to use ... for the design and the construction of the station " in the Page 5, Rows 47-48
- 4) The cross-references to Fig. 9a,b,c,d,e are not correct. I think these references are related to Fig. 6 in the Page 7.
- 5) "he overburden pressure" should be "the overburden pressure" in the Page 8, Row 3.
- 6) Page 16, Row 29: of STS structure has s high stiffness? Maybe it should be "a".
- 7) "flange flanges" in the Page 23 Row 3?

Author's Response to Decision Letter for (RSOS-181035.R0)

See Appendix A.

RSOS-181035.R1 (Revision)

Review form: Reviewer 1

Is the manuscript scientifically sound in its present form?

Yes

Are the interpretations and conclusions justified by the results?

Yes

Is the language acceptable?

Yes

Is it clear how to access all supporting data?

Yes

Do you have any ethical concerns with this paper?

No

Have you any concerns about statistical analyses in this paper?

No

Recommendation?

Accept as is

Comments to the Author(s)

The manuscript has been revised according to my comments, now I have no question.

Review form: Reviewer 3

Is the manuscript scientifically sound in its present form?

Yes

Are the interpretations and conclusions justified by the results?

Yes

Is the language acceptable?

Yes

Is it clear how to access all supporting data?

Yes

Do you have any ethical concerns with this paper?

No

Have you any concerns about statistical analyses in this paper?

No

Recommendation?

Accept as is

Comments to the Author(s)

Thank you for your response. The authors addressed my remarks and concerns.

Decision letter (RSOS-181035.R1)

02-Nov-2018

Dear Dr Jia:

On behalf of the Editors, I am pleased to inform you that your Manuscript RSOS-181035.R1 entitled "Study on surface settlement and structural deformation for large span subway station using a new pre-supporting system" has been accepted for publication in Royal Society Open Science subject to minor revision.

The referees are broadly satisfied that your paper is ready for publication; however, the Editors remain concerned the language is not yet ready for acceptance. Please can you seek advice from an appropriate English language polishing service (<https://royalsociety.org/journals/authors/language-polishing/>), and resubmit your paper, with evidence that you have sought advice. Thanks in advance for your revision.

Because the schedule for publication is very tight, it is a condition of publication that you submit the revised version of your manuscript before 11-Nov-2018. Please note that the revision deadline will expire at 00.00am on this date. If you do not think you will be able to meet this date please let me know immediately.

Please note that Royal Society Open Science charge article processing charges for all new submissions that are accepted for publication. Charges will also apply to papers transferred to Royal Society Open Science from other Royal Society Publishing journals, as well as papers submitted as part of our collaboration with the Royal Society of Chemistry (<http://rsos.royalsocietypublishing.org/chemistry>). If your manuscript is newly submitted and subsequently accepted for publication, you will be asked to pay the article processing charge, unless you request a waiver and this is approved by Royal Society Publishing. You can find out more about the charges at <http://rsos.royalsocietypublishing.org/page/charges>. Should you have any queries, please contact openscience@royalsociety.org.

Kind regards,
Royal Society Open Science Editorial Office

on behalf of Prof R. Kerry Rowe (Subject Editor)
openscience@royalsociety.org

Reviewer comments to Author:
Reviewer: 3

Comments to the Author(s)
Thank you for your response. The authors addressed my remarks and concerns.

Reviewer: 1

Comments to the Author(s)
The manuscript has been revised according to my comments, now I have no question.

Author's Response to Decision Letter for (RSOS-181035.R1)

See Appendix B.

Decision letter (RSOS-181035.R2)

09-Jan-2019

Dear Dr Jia,

I am pleased to inform you that your manuscript entitled "Study on surface settlement and structural deformation for large span subway station using a new pre-supporting system" is now accepted for publication in Royal Society Open Science.

on behalf of Prof R. Kerry Rowe (Subject Editor)
openscience@royalsociety.org

Appendix A

Dear Editors and Reviewers

Thanks for your careful review and your comments with regard to my manuscript: “A Case Study on the Application of the Steel Tube Slab Structure in Construction of a Subway Station” (Manuscript ID: RSOS-181035).

The comments are beneficial for me to revise and improve the manuscript. The manuscript has been revised according to the comments. The responds to the reviewers are as follows:

Reviewer1#

This paper describes an interesting pre-supporting system, called Steel Tube Slab (STS), that can be useful especially in case of large span underground excavation close to ground surface. Based on the numerical simulation and in situ monitoring, STS structure seems to be very effective and dependable measure in controlling and reducing the surface settlement and the existing surrounding buildings. Moreover, an interesting section of the paper is related to the influence of the excavation step length on the horizontal displacements at the tunnel face and on the ground settlement. I think that It's very useful for the application and promotion of STS technique. However, In order to improve the level of this manuscript better, some revisions need to be done, such as the following.

1. “I suggest the Authors highlighting/mentioning the main findings obtained at the end of the introduction section. “Several key parameters in the design ...” is too generic.”

Answer: Several key parameters have been produced in the revised manuscript in detail. They are as follows: The effect excavation process on ground settlement, deformation of STS structure and bridge pile are studied, moreover the key parameter such as welding of flanges and spacing of temporary steel supports are investigated using numerical simulations. Moreover, the other parts of the introduction have been perfected.

2. The problem studied by the Authors is a very complex soil-structure interaction problem, and I have appreciated the workflow of the paper, nevertheless I think that the soil conditions at the site are in general poorly described. So, if it is possible I recommend providing 1 Figure.

Answer: The soil profile has added, Figure6. And depth of the water level is from 5.2m to 14.9m in the construction area.

Figure 6 geological profile of the soil

3. Section 5.2. I clearly understand the complexity of such a numerical model, however why did the Authors select the Mohr-Coulomb model? It can be considered sufficient for this case-study?

Answer: Firstly, the relative parameters of Mohr-Coulomb model are more easily obtained. Meanwhile, it is a common phenomenon that M-C model is used to calculate the complex construction sequences of underground space, and has obtained great application results.

4. Fig. 20 - Why are used 0.7 m long element in simplified model of STS structure.

Answer: The connection of adjacent steel tubes is shown in below the Fig.1, from the figure, we can see the distance between the top flange and bottom flange is 0.7m, and the diameter of the steel tube is 0.9m. Considering the construction security, in the process of equivalent calculation, the thickness of the simplified model of STS structure is used 0.7m, which is smaller than actual thickness. In order to acquire the conservative computing data. Subsequently when the data is used to instruct the construction, they can ensure the construction security better.

Fig. 1 Connection of adjacent steel tubes

5. Fig.24. the numerical solutions presented in Fig. 24 are direct predictions or the result of a back-analysis.

Answer: Firstly, Comparison of surface settlement between in-suit monitoring and numerical simulations is investigated to verify the rationality of parameters selection, subsequently based on the numerical model, the effect excavation process on ground settlement, deformation of STS structure and bridge pile are studied, moreover the key parameter such as welding of flanges and the step length are studied.

6. Page 2, Line 12. "in-suit" should be "in-situ...". This error is repeated several times in all the paper;

Answer: It has been revised in the manuscript.

7. Page 5, Line 54. "statin" should be "station";

Answer: It has been revised in the manuscript.

8. Page 8, Line 15. Please check the value of 0.022mm, I think it is not correct. Probably should be 16 mm.

Answer: It has been revised in the manuscript.

9. Page 15, Lines 31-32. *"It indicate that ... were validity" I think it should be better to change with "It indicates that ... were valid";*

Answer: It has been revised in the manuscript.

10. Page 16, Line 29. *"of STS structure has s high stiffness" should be "of STS structure has a high stiffness";*

Answer: It has been revised in the manuscript.

Reviewer2#

1. *The title is too broad, a new title "Study on the effect of combining the new supporting structure with PBA method for a large span subway underground station" is be suggested.*

Answer: The title has been revised, "Study on surface settlement and structural deformation for large span subway station using a new pre-supporting system".

2. *The relative literatures can be added in the introduction.*

Answer: The relative literatures have been perfected in the revised manuscript.

3. *Some spelling and grammatical errors need to be modified, such as page2 "in-suit", page5 "statin", etc. Check all the paper.*

Answer: they have been revised in the manuscript.

4. *Page 11: Usually the Puason coefficient is indicated by "v".*

Answer: It has been revised in the manuscript.

5. *In Fig.17, it is not clear what does the colored part represent.*

Answer: It has been revised in the manuscript.

6. *The length of the paper is a little long, and the length should be reduced properly.*

Answer: It has been revised in the manuscript.

Reviewer3#

STS construction method is first to be proposed, and according to the numerical calculation and the field monitoring, the authors obtain that STS structure can reduce and control settlements and reduce unpleasant effects on existing buildings in the neighborhoods of the excavation site. It is a good paper for the later engineering application, but I have some concerns/doubts about the manuscript and suggest the author make several minor versions.

Comments and Questions

1) Fig. 3 (left) is similar to Fig. 6 (f). on the other hand, Fig. 3 (right) is identical to Fig. 7. Please check that.

Answer: Yes, indeed, Fig. 3 (left) is similar to Fig.6 (f) and, Fig.3 (right) is identical to Fig. 7. Fig. 3 is used to introduce the proposed support system, Steel Tube Slab method. The engineering application is introduced in the Fig. 6 (f) and Fig. 7. In a word, they have been revised in the manuscript.

2) Table 2. How to get submitted E?

Answer: As for the elastic modulus E, top longitudinal beam, bottom longitudinal beam and cylindrical steel column are all composed of steel and concrete, so they are specified by the equation:

$$EA = E_s A_s + E_c A_c$$

$$EA = E_s A_s + E_c A_c$$

where, E- the equivalent elastic modulus; A- the total cross area; E_s, E_c - elastic modulus of steel and concrete, respectively; A_s, A_c – cross area of steel and concrete, respectively.

3) As for the numerical model, the relationship between soil and structure is a complex question, how did the Authors consider?

Answer: The interface elements are used to simulate the interaction between soil and structure, and the interface elements are comprised of a series of the triangular elements contained three nodes, as is shown in Figure 1. The normal and shear constitutive equation of interfaces elements are as follows:

$$F_n^{(t+\Delta t)} = k_n \mu_n A + \sigma_n A$$

$$F_{si}^{(t+\Delta t)} = F_{si}^{(t)} + k_s \Delta \mu_{si}^{(t+0.5\Delta t)} A + \sigma_{si} A$$

where, $F_n^{(t+\Delta t)}$ is the normal at time $(t+\Delta t)$; $F_{si}^{(t+\Delta t)}$ is the shear force vector at time $(t+\Delta t)$; μ_n is the absolute normal penetration of the interface node into the target face; μ_{si} is the incremental relative shear displacement vector; σ_n is the additional normal stress added due to interface stress initialization; σ_{si} is the additional shear stress added due to interface stress initialization; k_s is shear stiffness, k_n is normal stiffness, and A is representative area associated with the interface node.

Figure 1 Interface element

The absolute normal penetration of interface surfaces and contact nodes and shear relative velocity are calculated in every step. Subsequently, they are substitute into the constitutive equations of interface elements to obtain normal stress vector and shear stress vector.

4) *In section 5.3, there should have the construction history depending on the staged construction. The construction history will justify the heave, which is observed in Fig. 24.*

Answer: The blue points represent the actual settlement of the construction history in the figure, but the figure has been replaced in order to improve the quality of the manuscript.

5) *From the framework of this paper, it seems that the fourth chapter is not strictly necessary, I suggest the author delete the part to increase the continuity of this paper. If it is possible, the author can make some the initial parameter analysis in this chapter using the numerical software.*

Answer: The manuscript has been revised according to the comments, and the framework of the manuscript has been adjusted.

Typo and grammatical

1) *In suit? It should be "in situ", for example Page 2 Row 11, please check the text.*

Answer: It has been revised in the manuscript.

2) *"...a new technique for are presented..." should be "...a new technique for ... is presented" in the Page 5, Row 7.*

Answer: It has been revised in the manuscript.

3) *"it was decided ... , is used to design and construct the station" I think is better to change with "it was decided to use ... for the design and the construction of the station " in the Page 5, Rows 47-48*

Answer: It has been revised in the manuscript.

4) *The cross-references to Fig. 9a,b,c,d,e are not correct. I think these references are related to Fig. 6 in the Page 7.*

Answer: It has been revised in the manuscript.

5) *“he overburden pressure” should be “the overburden pressure” in the Page 8, Row 3.*

Answer: It has been revised in the manuscript.

6) *Page 16, Row 29: of STS structure has s high stiffness? Maybe it should be “a”.*

Answer: It has been revised in the manuscript.

7) *“flange flanges” in the Page 23 Row 3?*

Answer: It has been revised in the manuscript.

Appendix B

Dear Editors

Thanks for your careful review and your comments with regard to my manuscript: "Study on ground settlement and structural deformation for large span subway station using a new pre-supporting system" (Manuscript ID: RSOS-181035.R2).

As for the reviewers, there is no question for the revised manuscript in the last email, so I do not provide a response to reviewers document which responds to each of the reviewers' comments point by point. But the language of the manuscript has been revised and improved according to the comments, and the modified part has been highlighted in the revised manuscript.